

# Theacrine alleviates sepsis-induced acute kidney injury by repressing the activation of NLRP3/Caspase-1 inflammasome

Maoxian Yang[1], Peng Shen[1], Longsheng Xu[2], Min Kong[3], Congcong Yu[4] and Yunchao Shi[1]

[1] Department of Intensive Care Unit, The First Hospital of Jiaxing, Affiliated Hospital of Jiaxing University, Jiaxing, China

[2] Department of Center Laboratory, The First Hospital of Jiaxing, Affiliated Hospital of Jiaxing University, Jiaxing, China

[3] Department of Anesthesiology, The First Hospital of Jiaxing, Affiliated Hospital of Jiaxing University, Jiaxing, China

[4] Department of Pharmacy, The First Hospital of Jiaxing, Affiliated Hospital of Jiaxing University, Jiaxing, China

Corresponding author
Yunchao Shi, luleisyc@163.com

## ABSTRACT

Acute kidney injury (AKI) is a frequent and serious complication of sepsis, which results in a rapid decline of kidney function. Currently, there are no curative therapies for AKI. Theacrine is a purine alkaloid and exerts significant role in regulating inflammation, oxidative stress, and mood elevation. The study aims to evaluate the biological role and potential mechanism of theacrine in septic AKI. The murine and cellular models of septic AKI were established in lipopolysaccharide (LPS)-treated C57BL/6 mice and HK-2 cells, respectively. The effect of theacrine on alleviating septic AKI was assessed after pretreatment with theacrine *in vivo* and *in vitro*. We found that theacrine treatment significantly alleviated LPS-induced kidney injury, as evidenced by decreased levels of kidney injury markers (blood urea nitrogen and creatinine), inflammatory factors (IL-1$\beta$ and IL-18), and cell apoptosis *in vivo* and *in vitro*. Mechanistically, theacrine markedly repressed the activation of NOD-like receptor (NLR) pyrin domain-containing protein 3 (NLRP3)inflammasome. As expected, MCC950 (a specific inhibitor of NLRP3) treatment also decreased LPS-induced production of IL-18 and IL-1$\beta$ and cell apoptosis in HK-2 cells. More important, Nigericin sodiumsalt (a NLRP3 agonist) damaged the effect of theacrine on repressing kidney injury markers (blood urea nitrogen and creatinine), pro-inflammatory cytokines (IL-18 and IL-1$\beta$), and cell apoptosis. Taken together, these results demonstrate that theacrine alleviates septic AKI, at least in part by repressing the activation of NLRP3 inflammasome.

## INTRODUCTION

Sepsis is a clinically complicated syndrome caused by infection, characterized by systemic inflammation after infection and extensive tissue damage (*Li et al., 2020*; *Yegenaga et al., 2004*). AKI is one of the common and severe complications in septic patients and occurs in approximately 30% of such patients (*Bellomo et al., 2017*). AKI is also commonly present in patients with multiple organ dysfunction. The severe sepsis-induced AKI (septic AKI)

is correlated with mortality of 50–70% (*Uchino et al., 2005*; *Wald et al., 2009*), and the survivors of septic AKI exhibit an elevated incidence of chronic renal disease (*Mayeux & MacMillan-Crow, 2012*). For septic AKI, there is currently no effective treatment. Therefore, revealing the underlying pathogenesis of septic AKI is urgent to effectively treat the disease.

Theacrine, a natural purine alkaloid extracted from *Camellia assamica* var. *kucha* (also known as *Camellia kucha* Hung T. Chang), possesses an anti-inflammation and analgesic effects (*Wang et al., 2010*). Like caffeine, theacrine exhibits significant anti-oxidant activity. *Zhou et al. (2019)* demonstrated that theacrine contributes to improve intervertebral disc degeneration through activating sirtuin 3/FOXO3/SOD2 signaling and attenuating oxidant stress. Theacrine possesses the anti-oxidant capacity by increasing the levels of catalase, superoxide dismutase (SOD), and glutathione peroxidase (GSH-Px) (*Li et al., 2013*).

Emerging studies showed that theacrine has a beneficial effect on maintaining brain function. *Samii, Nutt & Ransom (2004)* demonstrated that theacrine improves central fatigue-induced learning and memory impairment. Treatment of theacrine plus caffeine contributes to improve cognitive performance (*Ziegenfuss et al., 2017*). Theacrine also alleviates restraint stress-induced liver injury in mice (*Li et al., 2013*). Theacrine exhibits an anti-inflammatory effect by impairing the function of pro-inflammatory mediators including serotonin, bradykinin, and histamine (*Wang et al., 2010*). The anti-inflammatory effect of theacrine is associated with the reduction of capillary permeability and the increase of transforming growth factor-$\beta$ (TGF-$\beta$)-mediated shifts in the expression of pro-inflammatory cytokines (*Gao et al., 2020*). The alterations of deleterious inflammatory cascade are consistently existed in septic AKI regardless of disease severity or stage, suggesting the important role of these factors in the pathogenesis of septic AKI (*Mir et al., 2018*; *Takasu et al., 2013*). However, the effect of theacrine on regulating septic AKI remains unknown.

During sepsis, inflammatory cascade is usually regulated by NLRP3 inflammasome (*Huang et al., 2020*; *Mao et al., 2013*). Inflammasome is a multi-protein complex that controls caspase-1-dependent inflammation and cell survival. NLRP3 is the most well-characterized member of NLR inflammasome family, and consists of NLRP3, procaspase-1, and apoptosis-associated speck-like protein (ASC) (*Wang et al., 2020*). NLRP3 interacts with ASC and activates pro-caspase-1 to generate active caspase-1, and results in a subsequent production of mature IL-1$\beta$ and IL-18 (*Lu et al., 2014*; *Mangan et al., 2018*). NLRP3 inflammasome is activated in septic AKI, and inhibition of NLRP3 inflammasome effectively prevents disease progression (*Gong et al., 2015*; *Mao et al., 2013*). At present, the regulatory role of theacrine in NLRP3 inflammasome in septic AKI is unknown. Based on these findings, we explored whether theacrine protects against septic AKI by inhibiting NLRP3 inflammasome activation.

## MATERIALS AND METHODS

### Animals

Eighteen C57BL/6 mice (male, 6–7 weeks-old) were obtained from Charles River Laboratories (Beijing, China) and maintained in a controlled environment (temperature:

**Figure 1 The structure of theacrine.**

21−23 °C, humidity: 40–60%) in polypropylene cages (three mice per cage) with 12 h light-dark cycle. The experimental protocol was approved by the Animal Ethics Committee of the Jiaxing University (No. JUMC2021-031). Mice were allowed free access to standard rodent diet and drinking water, and treated in compliance with the guideline of Animal Ethics Committee of the Jiaxing University to reduce their suffering. Mice were randomly assigned into 3 groups after one-week. Six mice were treated with sterile saline for 24 h, six mice were treated with LPS (10 mg/kg) for 24 h, and six mice were pretreated with Theacrine (20 mg/kg, Fig. 1) (*Zheng et al., 2002*) for 12 h in the presence or absence of Nigericin sodium salt (NSS, 20 mg/kg) and then treated with LPS (10 mg/kg) for 24 h. Health of each mouse was observed during the experiment. All mice were given drugs with an intraperitoneal injection and euthanized through isoflurane anesthesia. Before the end of experiment, any animal showing persistent self-harm behavior or signs of unexpected disease will be euthanized immediately.

## Cell culture

HK-2 cells were obtained from Bluefbio (Shanghai, China) and cultured in Dulbecco's Modified Eagle's medium (DMEM; Sigma, MO, USA) supplemented with 10% FBS (GIBCO) and 1% penicillin/streptomycin in 5% $CO_2$ incubator at 37 °C. HK-2 cells were stimulated with different doses of LPS (0, 1, 2, 4, 8, and 10 μg/ml) for 24 h, or treated with 2 μg/ml of LPS for different time (0, 8, 12, 24 and 36h). The control group of cells were treated with PBS. At several experiments, HK-2 cells were treated with 2 μg/ml of LPS in the presence of Theacrine (1 μM; Merck, Kenilworth, NJ, USA), MCC950 (10 μM; Merck, Princeton, NJ, USA), or Nigericin (MedChem Express, 10 μM).

## Terminal deoxynucleotidyl transferase dUTP nick end labeling (TUNEL) assay

Renal tissue sections were dewaxed and rehydrated through xylene/ethanol, then digested with proteinase K (15 µg/ml; Beyotime, Shanghai, China) treatment for 20 min at 37 °C. After that, sections were stained using TUNEL kit according to the manufacturer's instructions (MBL, Beijing, China). *In vitro*, HK-2 cells were treated with 2 µg/ml of LPS in the presence of Theacrine (1µM) or MCC950 (10 µM), and then were treated with Immunol Staining Fix Solution (Beyotime) for 35 min and stained with TUNEL (green). DAPI (Beyotime) was applied to stain cell nuclei (blue). Sections and cells were observed with fluorescence microscopy (Caikon DFM-60; Caikon, Shanghai, China) at 200x magnification.

## Hematoxylin-eosin (HE) staining

The renal tissues were fixed with 4% paraformaldehyde for 24 h at 4 °C, dehydrated with ethanol and embedded in paraffin. The samples were cut into 6 µm thick sections by a microtome (Thermo Fisher, Waltham, MA, USA). Paraffin sections were stained with HE (Beyotime), and then sections were observed by the optical microscope (Leica DM1000; Leica, Wetzlar. Germany).

## Cell viability assay

HK-2 cell viability was assayed using Cell Counting Kit-8 (CCK-8, Beyotime). HK-2 cells ($4 \times 10^3$) were seeded in 96-well plates, treated with different doses of LPS for 24 h or treated with 2 µg/ml of LPS for different time. CCK-8 (10 µL/well) solution was added into each well and incubated in a CO2 incubator. Four hours later, the absorbance was measured at 450nm using a microplate reader (Detie HBS-ScanX; Detie, Nanjing, China).

## Quantitative real-time PCR (qPCR)

Total RNA was extracted from HK-2 cells using Trizol reagent (Sigma-Aldrich). The AMV reverse transcriptase (EMD Millipore, Billerica, MA, USA) was used to synthesize the cDNA based on the manufacturer's recommendations. qPCR was carried out by SYBR Green PCR Master Mix (APExBIO, Houston, TX, USA) on an CFX96 Touch Real-Time PCR System (Bio-Rad, Hercules, CA, USA) at 95 °C for 45 s, 40 cycles of 95 °C for 10 s, and 60 °C for 20 s. $\beta$-actin was used as internal control. Relative mRNA level was calculated through the $2^{-\Delta\Delta CT}$ method. qPCR primers used in the study were: IL-1$\beta$ forward, 5′-CTCGCAGCAGCACATCAAC-3′, IL-1$\beta$ reverse, 5′-AAGGTCCACGGGAAAGACAC-3′; IL-18 forward, 5′-TTGACAACACGCTTTACTTTATACC-3′, IL-18 reverse, 5′-AGTCTGGTCTGGGGGTTCACTG-3′; NLRP3 forward, 5′-AGTGGATGGGTTTGCTGGGAT-3′, NLRP3 reverse, 5′-TGCGTGTAGCGACTGTTGAGG-3′. $\beta$-actin forward, AGAGGGA AATCGTGCGTGAC, $\beta$-actin reverse, ATACCCAAGAAGGAAGGCTGG.

## Western blot analysis

Total protein was prepared from HK-2 cells in RIPA buffer (Beyotime) and quantified by BCA Protein Assay Kit (Abcam, Cambridge, UK)) based on the manufacturer's recommendations. Equal amounts of proteins (approximately 10 µg) were separated

via 10% SDS-PAGE and then transferred onto polyvinylidene fluoride (PVDF) membranes (Thermo Fisher). After blocking with five percent bovine serum albumin (5% BSA) at 37 °C for 1 h, membranes were incubated with NLRP3 (1:1000, ab263899; Abcam), Cleaved-Cas-1 (1:1000, D57A2; Cell Signaling Technology), Caspase-1 (1:1000, ab207802; Abcam) and $\beta$-actin (1:5000, ab6276; Abcam) overnight at 4 °C. Then membranes were incubated with HRP-conjugated goat anti-mice IgG (1:10000, ab6789; Abcam) and goat anti-rabbit IgG (1:50000, ab205718, Abcam) for 1.5 h at 37 °C. The band intensities were analyzed using an ECL kit (Thermo Fisher) and quantified with NIH ImageJ software (Bethesda, MA, USA).

### Analysis of kidney function

Blood samples were acquired from all mice to evaluate kidney function before euthanasia (*Liu et al., 2020b*). Serum levels of BUN (blood urea nitrogen) and SCR (serum creatinine) were assayed using commercial enzyme linked immunosorbent assay (ELISA) kits (Beyotime).

### Statistical analyses

Data are presented as means ± standard error of mean (SEM) from independent repetitions. Statistical analyses were performed using ANOVA or unpaired two-tailed Student's $t$-test when comparing multiple groups with GraphPad Prism 7.0 (GraphPad, CA, USA). $P < 0.05$ was considered to be statistically significant.

## RESULTS

### Theacrine alleviated septic AKI, pro-inflammatory response, and cell apoptosis *in vivo*

To evaluate the effect of theacrine on the pathogenesis of septic AKI, C57BL/6 mice were pretreated with theacrine via injected intraperitoneally for 12 h and then administered intraperitoneally with LPS for 24 h. As shown in Fig. 2A and 2B, mice treated with LPS showed increased levels of BUN and creatinine, whereas theacrine treatment effectively reversed the effect. The results from H&E staining revealed that LPS administration resulted in renal histopathological damages, including severe interstitial edema, inflammatory cell infiltration, and vacuolar degeneration, whereas theacrine decreased these damages (Fig. 2C). Figures 2D and 2E showed that theacrine treatment significantly repressed LPS-induced pro-inflammatory cytokines, such as IL-18 and IL-1$\beta$. To acquire more evidence verifying the role of theacrine in alleviating septic AKI, the effect of theacrine on repressing LPS-induced cell apoptosis was assessed using TUNEL staining, and data showed that theacrine significantly decreased the number of apoptotic cells in the kidney tissues from LPS-treated mice (Fig. 2F and 2G). These results demonstrate that the mice model of septic AKI was successfully established, and theacrine pretreatment alleviates septic AKI *in vivo*.

### Theacrine repressed LPS-induced inflammatory response, cell apoptosis, and the activation of NLRP3 inflammasome *in vitro*

To establish the cellular model of septic AKI, human renal tubular epithelial cells (HK-2) were stimulated with different doses of LPS and then cell viability was assessed using

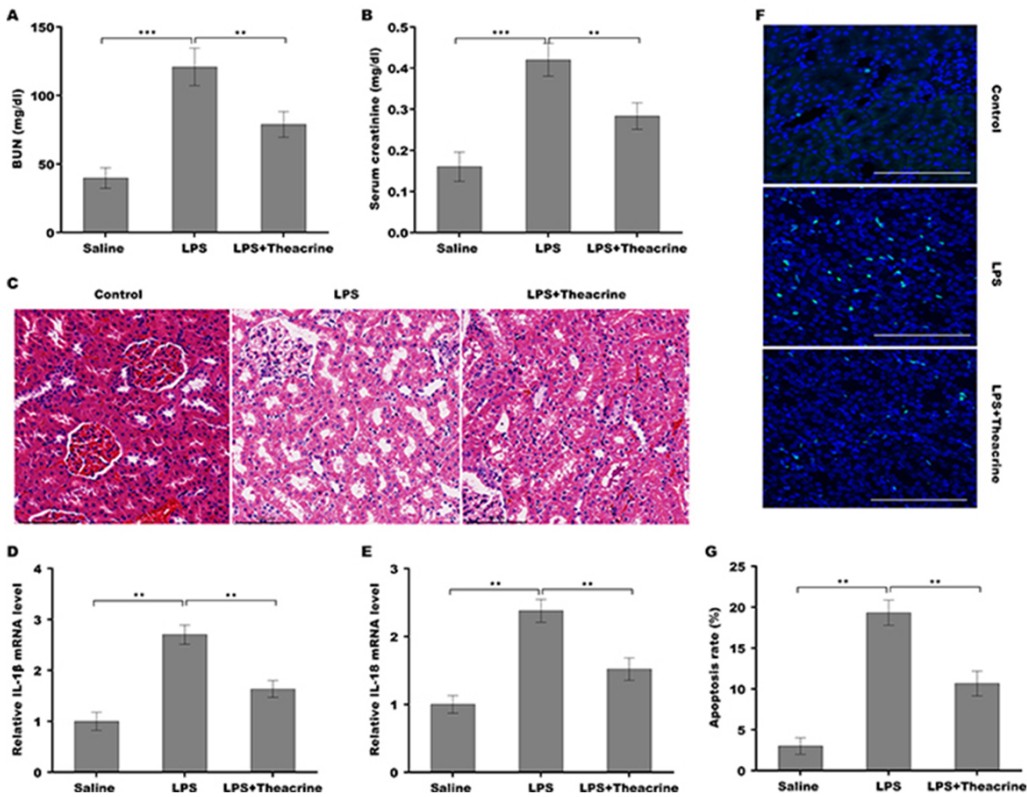

**Figure 2** **Theacrine alleviated septic AKI, pro-inflammatory response, and cell apoptosis *in vivo*.** The BUN (A) and SCR (B) levels were assessed using commercial ELISA kits in mice ($n = 6$ per group) administered intraperitoneally with LPS (10 mg/kg) for 24 h with or without pretreatment with theacrine (20 mg/kg) for 12 h. (C) HE staining of renal tissues in mice treated with LPS (10 mg/kg) for 24 h with or without pretreatment with theacrine (20 mg/kg) for 12 h. (D and E) qPCR analysis of IL-1$\beta$ and IL-18 mRNA levels in renal tissues. (F) TUNEL analysis in renal tissues in different groups. (G) Statistical analysis of results showed in (F). **$p < 0.01$.

CCK-8 reagent. Figure 3A showed that HK-2 cell viability was gradually reduced with increasing concentration of LPS, and was reduced markedly once the concentration of LPS reached 2 μg/ml, indicating that 2 μg/ml was an optimum dose of LPS to establish cellular model of septic AKI. We further assessed the effect of LPS on HK-2 cell viability at different time points, and Fig. 3B revealed that HK-2 cell viability was significantly reduced in a time-dependent manner. The role of theacrine in LPS-induced inflammation and apoptosis was next assayed in HK-2 cells. As shown in Fig. 3C, cell viability was markedly decreased after LPS treatment, whereas theacrine markedly reversed the effect. TUNEL staining analysis showed that theacrine decreased LPS-induced cell apoptosis (Fig. 3D and 3E). Furthermore, the mRNA levels of IL-18 and IL-1$\beta$ were significantly increased in HK-2 cells after LPS treatment, whereas theacrine treatment reversed the effect (Fig. 3F and 3G).

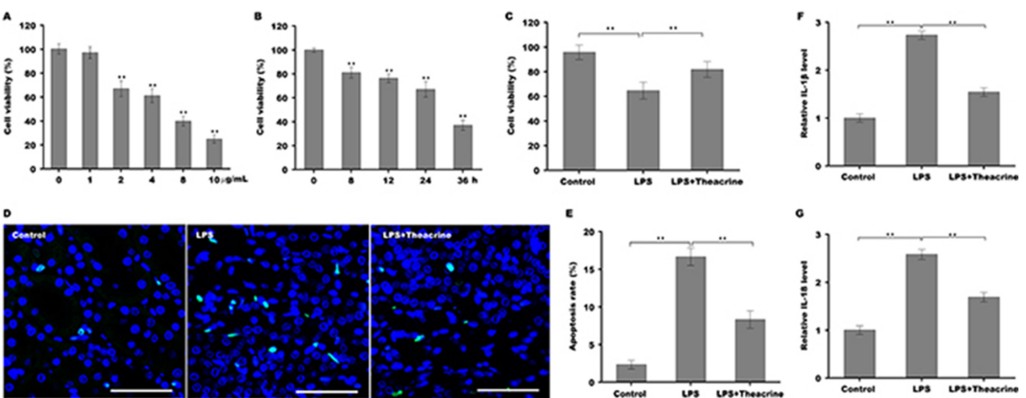

**Figure 3** **Theacrine repressed LPS-induced cell apoptosis and inflammatory response.** (A) HK-2 cell viability was assayed using CCK-8 reagent after treatment with different doses of LPS (0, 1, 2, 4, 8, and 10 μg/mL) for 24 h. (B) HK-2 cell viability was assayed using CCK-8 reagent after treatment with LPS (2 μg/mL) for 0, 8, 12, 24, and 36 h. (C) HK-2 cell viability was assayed using CCK-8 reagent after treatment with LPS (2 μg/mL) for 24 h in the presence or absence of Theacrine (1 μM). (D and E) HK-2 cell apoptosis was assayed using TUNEL staining after treatment with LPS (2 μg/mL) for 24 h in the presence or absence of Theacrine (1 μM). qPCR analysis of IL-$\beta$ (F) and IL-18 (G) levels in HK-2 cells after treatment with LPS (2 μg/mL) for 24 h in the presence or absence of Theacrine (1 μM). Each experiment was confirmed by three independent experiments. **$p < 0.01$.

## Theacrine repressed the activation of NLRP3 inflammasome

Given the regulatory role of theacrine in Sirtuin-3 (SIRT3) activation and the correlation of SIRT3 with NLRP3 inflammasome, we thus investigated whether theacrine regulated the activation of NLRP3 inflammasome. As shown in Figs. 4A–4C, LPS treatment resulted in marked increase of NLRP3 mRNA and protein levels, whereas theacrine repressed LPS-induced upregulation of NLRP3. In addition, theacrine also decreased LPS-induced increase of cleaved Caspase-1 expression (Fig. 4B and 4D), indicating theacrine repressed LPS-induced activation of NLRP3 inflammasome. As expected, MCC950, a specific inhibitor of NLRP3, also repressed LPS-induced production of IL-18 and IL-1$\beta$ (Fig. 5A and 5B), and cell apoptosis (Fig. 5C).

## Theacrine alleviated septic AKI through repressing the activation of NLRP3 inflammasome

Finally, we explored whether theacrine alleviated septic AKI through repressing the activation of NLRP3 inflammasome *in vivo*. To this end, a specific activator of NLRP3 (Nigericin sodiumsalt, NSS) was used to assess the mediated role of NLRP3 in alleviating septic AKI. C57BL/6 mice were pretreated with theacrine in the presence or absence of NSS, and then injected with LPS. As shown in Fig. 6A and 6B, mice pretreated with theacrine plus NSS showed significant increase in BUN and creatinine levels compared with theacrine alone pretreatment. Mice pretreated with theacrine plus NSS also showed significantly increased production of IL-18 and IL-1$\beta$ compared with theacrine alone pretreatment (Fig. 6C and 6D). The results from TUNEL staining showed that the function of theacrine

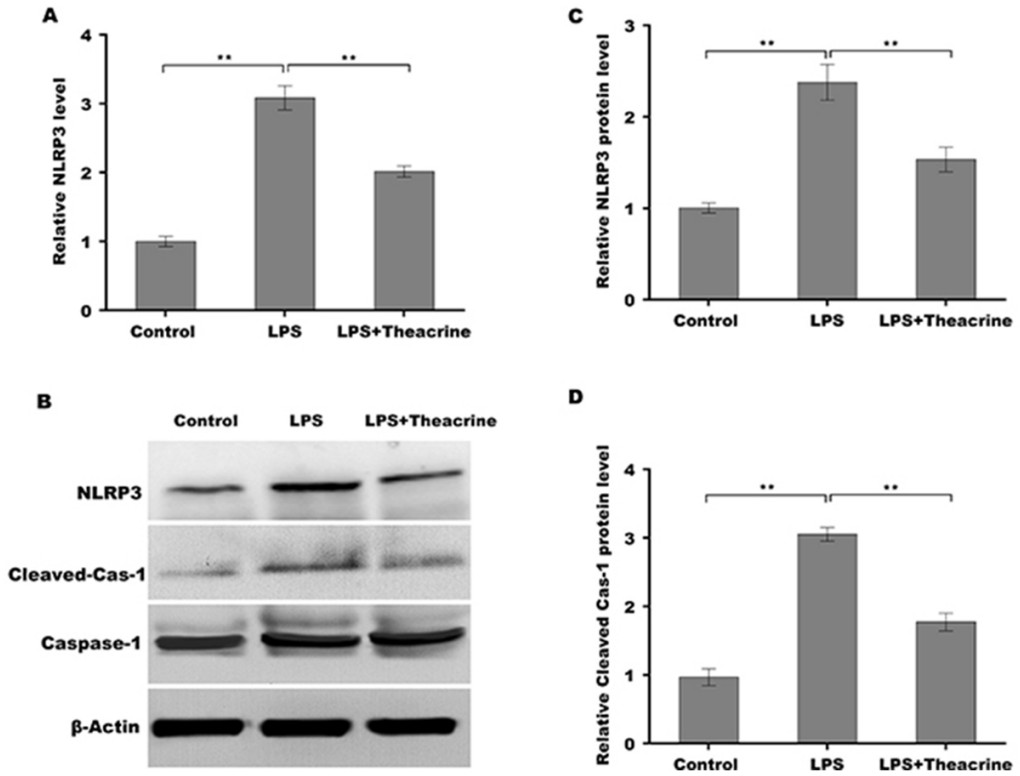

**Figure 4** **Theacrine repressed LPS-induced activation of NLRP3 inflammasome *in vitro*.** (A) qPCR analysis of NLRP3 mRNA level in HK-2 cells after treatment with LPS (2 μg/mL) for 24 h in the presence or absence of Theacrine (1 μM). (B) Western blot analysis of NLRP3 protein expression in HK-2 cells after treatment with LPS (2 μg/mL) for 24 h in the presence or absence of Theacrine (1 μM). Quantitative analysis of NLRP3 protein expression (C) and cleaved caspase-1 protein expression (D) showed in (B), Each experiment was confirmed by three independent experiments. $**p < 0.01$.

on alleviating LPS-AKI, such as cell apoptosis, were significantly reversed by NSS treatment (Fig. 6E).

# DISCUSSION

Tea possesses anti-microbial, anti-inflammatory, anti-oxidative, and neuroprotective effects (*Hayat et al., 2015*). Theacrine is a bioactive substance in *Camellia assamica* var. *kucha*, and has attracted more and more attention due to its multitudinous healthy benefits, such as anti-inflammation (*Gao et al., 2020*) and anti-oxidant effect (*Duan et al., 2020*), and facilitating mental clarity (*Ziegenfuss et al., 2017*). However, the effect of theacrine on septic AKI remains unknown. In the current study, we demonstrated that, (i) Theacrine decreased pro-inflammatory response and cell apoptosis, and alleviated septic AKI *in vivo*, (ii) Theacrine repressed LPS-induced activation of NLRP3 inflammasome and inflammatory response *in vitro*, (iii) Theacrine alleviated septic AKI through repressing NLRP3 inflammasome activation. These data reveal the role of theacrine in repressing

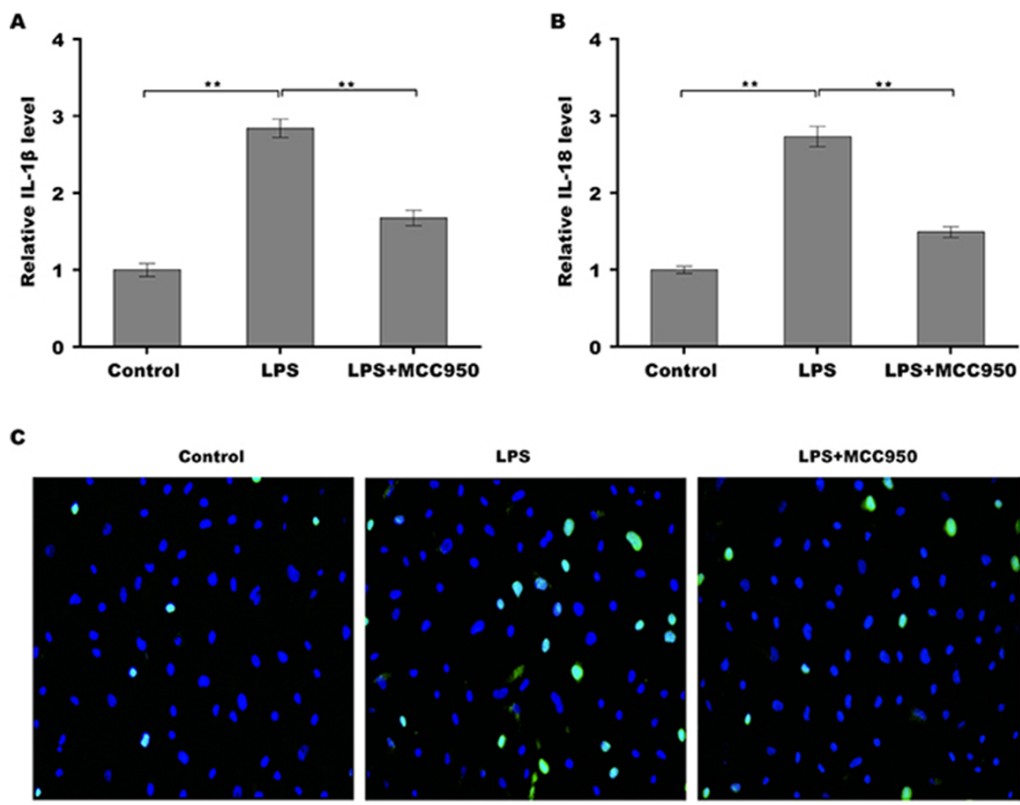

**Figure 5** **Theacrine repressed NLRP3 inflammasome activation.** qPCR analysis of IL-$\beta$ (A) and IL-18 (B) levels in HK-2 cells after treatment with LPS (2 $\mu$g/mL) for 24 h in the presence or absence of MCC950 (10 $\mu$M). (C) HK-2 cell apoptosis was assayed using TUNEL staining after treatment with LPS (2 $\mu$g/mL) for 24 h in the presence or absence of MCC950 (10 $\mu$M). Each experiment was confirmed by three independent experiments. **$p < 0.01$.

septic AKI, suggesting the potential of theacrine in the treatment of inflammation-related diseases.

Recent studies have demonstrated that abnormal activation of NLRP3 inflammasome is closely correlated with septic AKI (*Li et al., 2022a*; *Li et al., 2022b*; *Liu et al., 2020a*). miR-30c-5p is a key mediator of renal diseases and overexpression of miR-30c-5p attenuates septic AKI through targeting thioredoxin-interacting protein (TXNIP) and repressing NLRP3 activation (*Li et al., 2021*). *Wang et al. (2015)* reported that exogenous carbon monoxide (ECO) decreases oxidative stress and pro-inflammatory cytokines production, and thus reduces renal histology scores and increases survival rates in mice with septic AKI. They further demonstrated that ECO protects against the septic AKI through repressing the activation of NLRP3 inflammasome (*Wang et al., 2015*). *Huang et al. (2020)* found that pannexin-1 level is elevated in septic AKI mice and patients, and inhibition of pannexin-1 represses the generation of pro-inflammatory cytokines and cell apoptosis by suppressing NLRP3 inflammasome activation. Ginsenoside Rg1 treatment contributes to alleviate LPS-triggered kidney injury through repressing NLRP3 activation (*Zhang et al., 2022*).

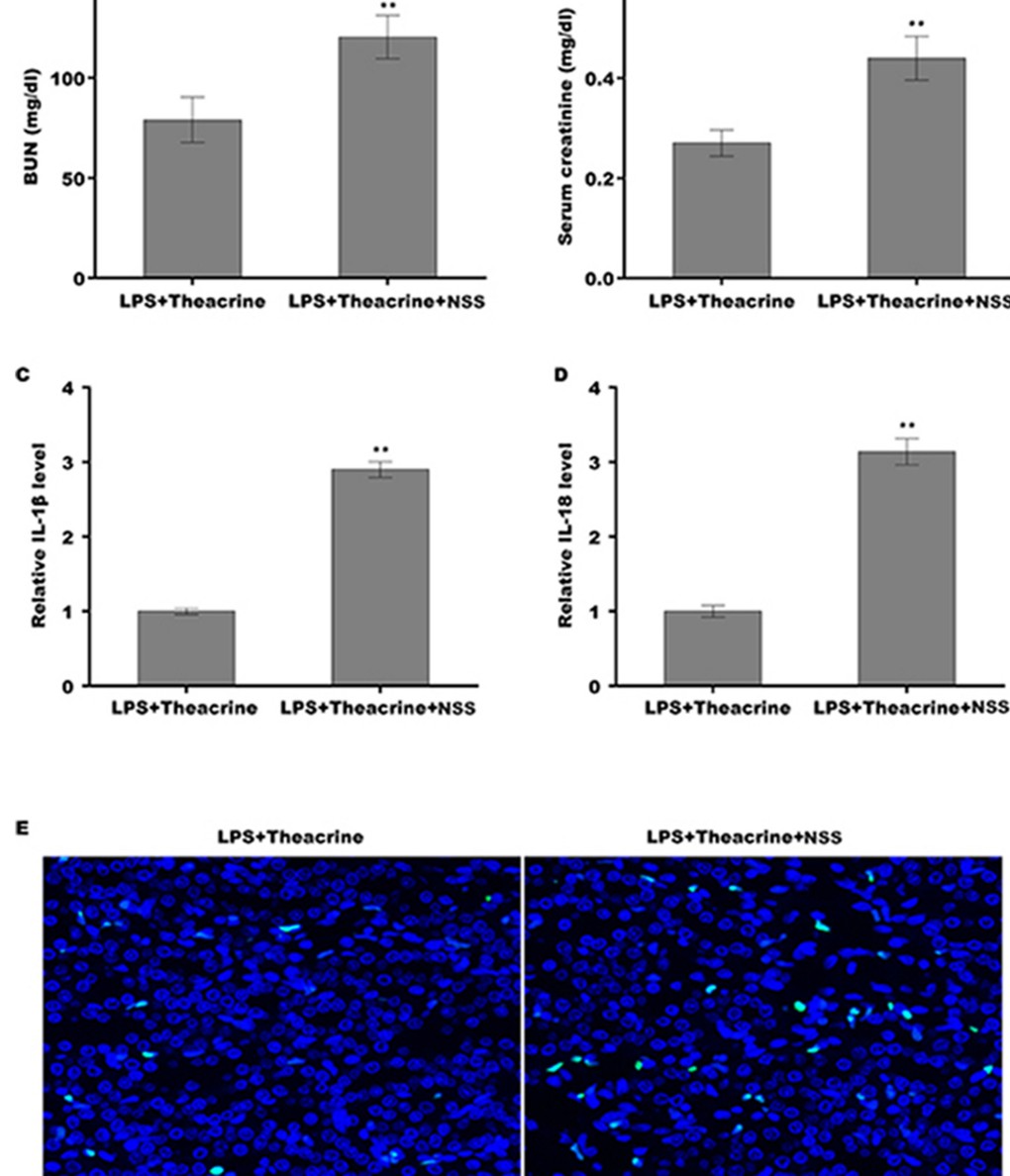

**Figure 6 Theacrine alleviated septic AKI through repressing NLRP3 inflammasome activation.** The levels of BUN (A) and SCR (B) were assessed using commercial ELISA kits in mice ($n = 6$ per group) administered intraperitoneally with LPS (10 mg/kg) plus Theacrine (20 mg/kg) in the presence or absence of Nigericin sodium salt (NSS, 20 mg/kg). qPCR analysis of IL-$\beta$ (C) and IL-18 (D) levels in mice administered with LPS (10 mg/kg) plus Theacrine (20 mg/kg) in the presence or absence of NSS (20 mg/kg). (E) TUNEL analysis in renal tissues in different groups. **$p < 0.01$.

These results indicate that NLRP3 inflammasome is a critical regulator of septic AKI, and it is necessary to inhibit NLRP3 inflammasome activation for alleviating septic AKI.

Although the regulatory role of theacrine in NLRP3 inflammasome is unclear, emerging studies revealed that theacrine decreases reactive oxygen species (ROS) accumulation via SIRT3-mediated deacetylation of superoxide dismutase 2 (SOD2) (*Duan et al., 2020*). Given the important role of ROS in initiating the assembly of NLRP3 with the ASC and subsequent pro-caspase-1 (*Biswas, 2016*; *Ramos-Tovar & Muriel, 2020*), we speculated whether theacrine exerts an anti-inflammatory role through regulating NLRP3 inflammasome. Here we demonstrated that LPS causes a marked increase of NLRP3 expression, whereas theacrine treatment reverses the process. As expected, theacrine reduces LPS-induced upregulation of cleaved Caspase-1, indicating that theacrine inhibits LPS-induced activation of NLRP3 inflammasome. Functionally, mice pretreated with theacrine plus NSS shows a significant increase in BUN and creatinine levels compared with theacrine alone treatment, suggesting that NSS-induced activation of NLRP3 inflammasome damages the effect of theacrine on protecting against septic AKI. Taken together, these data demonstrate that theacrine alleviates septic AKI by repressing NLRP3 inflammasome activation, indicating that theacrine has potential to treat septic AKI.

### Funding
This work was supported by the Jiaxing Science Technology Project (2018AY32017), the Key Subject of Jiaxing Medicine (2019-zc-12), and the Jiaxing Leading talent Plan (2019-lj-03). The funders had no role in study design, data collection and analysis, decision to publish, or preparation of the manuscript.

### Grant Disclosures
The following grant information was disclosed by the authors:
Jiaxing Science Technology Project: 2018AY32017.
Key Subject of Jiaxing Medicine: 2019-zc-12.
Jiaxing Leading talent Plan: 2019-lj-03.

### Competing Interests
The authors declare there are no competing interests.

### Author Contributions
- Maoxian Yang conceived and designed the experiments, performed the experiments, analyzed the data, prepared figures and/or tables, authored or reviewed drafts of the article, and approved the final draft.
- Peng Shen performed the experiments, prepared figures and/or tables, authored or reviewed drafts of the article, and approved the final draft.
- Longsheng Xu performed the experiments, analyzed the data, prepared figures and/or tables, and approved the final draft.

- Min Kong performed the experiments, prepared figures and/or tables, and approved the final draft.
- Congcong Yu performed the experiments, analyzed the data, prepared figures and/or tables, and approved the final draft.
- Yunchao Shi conceived and designed the experiments, performed the experiments, prepared figures and/or tables, authored or reviewed drafts of the article, and approved the final draft.

### Animal Ethics

The following information was supplied relating to ethical approvals (i.e., approving body and any reference numbers):

All experimental protocols were approved by the Animal Ethics Committee of the Jiaxing University(JUMC2021-031)

### Data Availability

The raw date are available in the Supplementary Files.

### Supplemental Information

Supplemental information for this article can be found online at http://dx.doi.org/10.7717/peerj.14109#supplemental-information.

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
