# Peer review of "Theacrine alleviates sepsis-induced acute kidney injury by repressing the activation of NLRP3/Caspase-1 inflammasome"

_PeerJ, doi:10.7717/peerj.14109_

## Round 0.1 · original submission · Major Revisions

The two expert reviewers gave thorough reviews and identified several key concerns. I urge you to address all of these concerns in your revised manuscript.

Reviewer 1 ·

Basic reporting

Basic reporting
1. The English used is acceptable, but could be improved.
Example Abstract:
there is no therapies – there are no therapies
after pretreatment with of theacrine
NLRP3 agonist damaged the effect

2. Literature references/background
It needs to be introduced that theacrine is found in tea. In the discussion the value of tea is mentioned, but the connection to theacrine is not made. What kind of tea, concentrations and some literature would support the case of theacrine use.
The introduction and discussion mention data on learning, memory, cognitive performance and liver injury. The authors need to explain more clearly why they suggest a role for theacrine in kidney injury/AKI.
Most important, as the role of inflammasomes is central in this manuscript, clearly more background on the role of inflammasome components in kidney disease is necessary. Especially the current knowledge about inflammasome components in human AKI and human septic AKI should be discussed using up to date literature to underline/explain an importance of the research data presented in this manuscript.
In general, it is preferable to find more than just one reference to support a statement.

3. Raw data. The Western blot raw data provided are not sufficient to support the quality of these data. The protein bands should not be presented as cut-out pieces and clearly a protein standard with kDa annotations should be visible.

Experimental design

Experimental design
4. The use of the TUNEL assay needs explanation to the reader.
a) explain the abbreviation
b) explain the principle and the conclusions that can be drawn from the experiments

5. please provide an exact description for all devices used. Example: Leica DM4 B & DM6 B upright digital research microscopes? Leica DM1000? Leica DM4 M & DM6 M? etc.

6. For the qPCR the beta-actin primers are missing. Was the annealing temperature for all primers identical?

7. For the Western blot: what was the protein loading?

8. Kidney function in mice: did the authors record urine volume or weight during the experiments?

9. In the results section and figures: For all statements about “expression” it should be specified, if gene expression or protein expression/protein amount was analyzed.

10. For all experiments (PCR, Western blot, TUNEL assay, tissue staining etc.) the number of experiments, animals, replicates etc. must be indicated. Example n=number of experiments in N=number of animals. For staining: representative example out of n= analyses.

Validity of the findings

11. The two major shortcomings in the experimental design are
a) The authors need to be exact about the difference between inflammasome expression, inflammasome priming and inflammasome activation. Changes in gene expression of IL-1beta and IL-18 only point to NFkappa-B involvement, which can contribute to the amount of mature IL-1beta and IL-18. Inflammasome activity/activation can only be concluded from an increase of mature IL-1beta and IL-18 protein in cells or after secretion/release.

b) The use of an NLRP3 activator (nigericin) does not really support the involvement of NLRP3 in the theacrine effect in AKI. The authors need to show the effect of any kind of inflammasome inhibition.

Reviewer 2 ·

Basic reporting

The manuscript by Maoxian Yang et al showed the effect of Theacrine in sepsis-induced AKI. The English in the manuscript is generally used well; however, an expert needs to revise again the grammar and spelling. The references are adequate and sufficient.
The article structure needs to improve. The results support the hypothesis

Experimental design

-The authors provide wb images, however they did not put which group each WB band belongs to. Please, add these data.
-The authors do not show all the N of the WB for the 6 per group.
- The changes they report in their graphs for figure 5C are not very evident, please provide better images. Please point out the changes with arrows on each representative image.
-Please mention in the figure caption the N you are using and what statistical test was applied.

Validity of the findings

The manuscript is novelty and the conclusions are well stated.

---

## Round 0.2 · Minor Revisions

The reviewer suggests that you address issues relating to current knowledge about inflammasome, and typographic errors.

Reviewer 1 ·

Basic reporting

Please find my comments on English language use and literature references below.

Experimental design

no comment

Validity of the findings

no comment

Additional comments

The manuscript is significantly improved. The methods' description and results presentation justify publication.

Nevertheless, there are two points that still need to be addressed:
1. Especially the current knowledge about inflammasome components in human AKI and human septic AKI should be discussed using up to date literature to underline/explain an importance of the research data presented in this manuscript. In general, it is preferable to find more than just one reference to support a statement.
I really want to stress, that the authors find relevant literature that assesses the topic in human AKI and human septic AKI (=patients). I do not assume that the authors' final aim of the research is the treatment of AKI/septic AKI in laboratory animals.

2. Please check again spelling and grammar, the revised paragraphs included.

---

## Round 0.3 · Minor Revisions

To finalize the acceptance of your manuscript, please address the issues stated below:

1. If it is not originally drawn, provide citation for the source of the structure of Theacrine in Figure 1.
2. Provide clarification for the definition of N= 3 in Figures 3, 4 and 5.
3. Provide in the supplementary information ALL replicate full gels from which cropped gel shown in Figure 4 was obtained.

We look forward to your next submission.

Reviewer 1 ·

Basic reporting

The basic reporting was improved, although I think that it is sad, that the authors claim that no clinical data on inflammasomes and AKI have been published.

Experimental design

no further comments

Validity of the findings

no further comments

Additional comments

none

Reviewer 2 ·

Basic reporting

The manuscript was improved and I recommend accept in the current form.

Experimental design

All experiments are realized adequately and the supplementary material support this.

Validity of the findings

The manuscript is novelty and the experiments support the obtained results.

---

## Round 0.4 · Minor Revisions

There are still inconsistencies in the descriptions of N= 3 (or n=3) in the captions to Figures 3, 4 and 5.
Figure 3: N=3 per group(Data are presented as means ± standard error of mean (SEM) from three independent repetitions.)
Figure 4: n=3 per group(Data are presented as means ± standard error of mean (SEM) from independent repetitions.)
Figure 5: n=3 per group(Data are presented as means ± standard error of mean (SEM) from independent repetitions.)
Is there any difference between N and n? Why is "three" missing in the description of n? Clarification must be provided to enable the readers properly evaluate the outcome of the study.

---

## Round 0.5 · accepted · Accept

We look forward to your next submission.